

# A closer look at four-dot masking of a foveated target

Marwan Daar and Hugh R. Wilson

Centre for Vision Research, York University, Toronto, Ontario, Canada

## ABSTRACT

Four-dot masking with a common onset mask was recently demonstrated in a fully attended and foveated target (*Filmer, Mattingley & Dux*, *2015*). Here, we replicate and extend this finding by directly comparing a four-dot mask with an annulus mask while probing masking as a function of mask duration, and target-mask separation. Our results suggest that while an annulus mask operates via spatially local contour interactions, a four-dot mask operates through spatially global mechanisms. We also measure how the visual system's representation of an oriented bar is impacted by a four-dot mask, and find that masking here does not degrade the precision of perceived targets, but instead appears to be driven exclusively by rendering the target completely invisible.

## INTRODUCTION

Over the last two decades, common onset masking with a four-dot mask (also referred to as object substitution masking) has proven to be a valuable way to study the processes by which a visual object is consciously rendered (*Goodhew et al.*, *2013*). In this masking paradigm, a target is briefly flashed along with four surrounding dots, the latter of which persist for a variable duration. As this mask duration increases, target visibility is reduced, a phenomenon that likely reflects competition between the target and mask. Given the sparse nature of this mask, relative to more traditional masks that either fully surround the target's contours or spatially camouflage it, four-dot masking is thought to be a powerful demonstration of interactions at a spatially global object level, rather than of spatially local contour interactions (*Di Lollo, Enns & Rensink*, *2000*). In addition to providing insights into the time-course of visual processing, four-dot masking offers a means to probe the way in which objects are individuated by the visual system (*Goodhew et al.*, *2015*; *Lleras & Moore*, *2003*). It also allows us to examine how attentional manipulations can bias competition between visual objects, such as a target and mask (*Pilling et al.*, *2014*; *Tata & Giaschi*, *2004*), and to explore the ability of a masked target to influence subsequent processing (*Choo & Franconeri*, *2010*; *Goodhew et al.*, *2011*).

Until recently, four-dot masking was only reliably reported when the target was presented in peripheral visual field, usually as part of a set of possible targets. Masking has also been found with a single target in peripheral visual field, but only reported in cases where the spatial location of this target was randomized (*Argyropoulos et al.*, *2013*; *Camp et al.*, *2015*).

Corresponding author
Marwan Daar,
marwan.daar@gmail.com

A recent report, however, demonstrates four-dot masking involving a fully attended target in central visual field (*Filmer, Mattingley & Dux, 2015*). This important finding clearly shows that neither distributed attention nor crowding are necessary conditions for this form of masking. The ability to study four-dot masking in central visual field with a single target is valuable for a few reasons. With multiple targets, there is the risk of noise introduced by pooling data across multiple target locations. With a single target location, one can therefore obtain well controlled data with relatively fewer trials. Furthermore, the use of a single, central target allows for efficient use of space within the visual field, and there are thus fewer limitations on parameters such as target size, and target-mask separation. Finally, the use of a single target offers the theoretical convenience of removing the influence of crowding, and feature misbinding (mistaking a feature of one of the distractors for a feature of the target), on any discovered effects.

In the current study, we sought to explore four-dot masking of a foveated target more closely. We were interested in two questions. First, how does a four-dot mask compare with an annulus mask, as a function of target-mask separation? As an annulus completely surrounds the target, it can mask the target through local inhibition as well as through object substitution/updating (*Enns, 2004*). It is unlikely, however, that a four-dot mask operates via local inhibition, given its sparse nature. Comparing these two mask types may provide evidence for a dissociation of these mechanisms. Second, we were interested in how this form of masking affects the target representation (*Agaoglu et al., 2015*; *Harrison, Rajsic & Wilson, 2016*). To do this, we developed a matching task where observers adjusted the orientation of a bar to match that of the target bar, and examined how the distribution of errors changed between baseline and masking conditions. This task also allowed us to examine whether the errors were more likely to occur when the target bar was closest to any of the four dots.[1]  If so, then this would point to the contribution of local masking mechanisms.

[1]We would like to thank Haluk Öğmen for providing the suggestion for this experiment.

## EXPERIMENT 1

Our first step was to see whether we could successfully replicate the masking effect found in *Filmer, Mattingley & Dux (2015)*. In their study, they used a forward noise mask in addition to a common onset trailing four-dot mask, and found a reliable performance drop as the duration of the four-dot mask increased.

### Methods & Procedures

The target was an annulus with a bar projecting from its centre to one of four cardinal points along the circumference. The inner radius of this annulus was 32.3 arcmin, and the outer radius was 39.5 arcmin. The projected bar, whose length was equal to the inner radius of the annulus, had a thickness of 5 arcmin. The mask comprised four small circles (diameter = 14.4 arcmin) located at the corners of an imaginary square concentric with the target. The separation of the mask, measured from the edge of each dot to the outer edge of the target annulus, was 31.6 arcmin. The forward mask was a circular patch of Gaussian noise, with a mean luminance equal to the that of the background, and whose radius was equal to the outer radius of the target annulus. A square cross of length 28.7 arcmin was
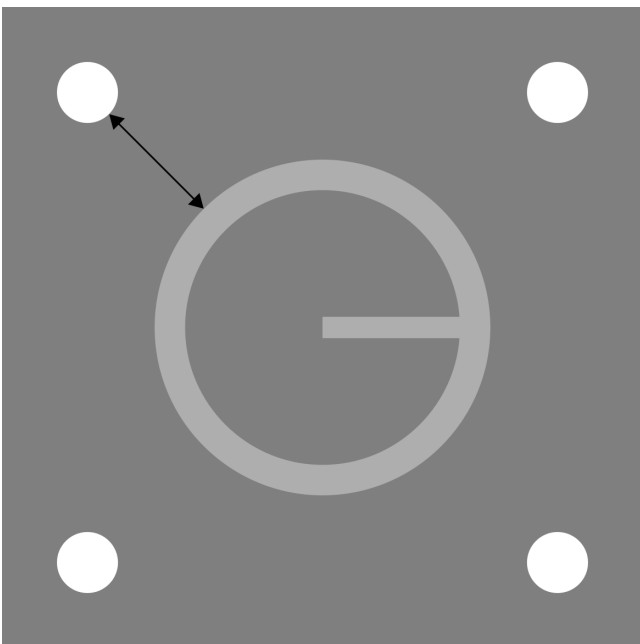

**Figure 1 Target and four-dot mask drawn to scale.** The black line indicates the distance along which target-mask separation was measured. The separation shown here was used in Experiment 1.

used for fixation. All stimuli were centrally presented with no spatial jitter. The target and four-dot mask are depicted in Fig. 1.

The luminance of the target, as measured with a Konica-Minolta LS-100 (integration time: 400 ms) was 65.8 cd/m$^2$, while that of the four-dot mask and fixation cross was 97 cd/m$^2$. All stimuli were presented against a uniform background grey field of luminance 47 cd/m$^2$, on a VIEWPixx display calibrated to linear light (gamma = 1), at a viewing distance of 1.28 m in a dimly lit room. The display was running at 120 hz in scanning backlight mode. In this mode, the backlight is scanned down the display in synchrony with the updated pixels, and the pixel rise time (black to white) and fall time (white to black) are both 1 ms. Thus, in a single frame comprising a white field, each pixel lets light through (above and beyond the black level luminance) for only 2 ms. It is important to precisely specify the mode in which visual content is rendered on the display, as this has implications for the actual stimulus durations, which are not always readily calculable based on reported frame rate (*Elze*, *2010*).

The trial sequence is shown in Fig. 2A. Each run was initiated with a keypress, at which point the fixation cross appeared for one second. As soon as the fixation disappeared, the forward mask appeared, lasting 200 ms. Immediately following its offset were the target and four-dot mask (henceforth termed "4DM"). The target persisted for 8.3 ms. In the common offset condition, which served as a baseline, the mask disappeared with the target (mask duration = 8.3 ms). Two other mask durations were tested: 250 ms, and 500 ms. Upon mask offset, the background remained visible, and a keypress response indicated which of the four directions the observer believed the line to be pointing along (up, down, left, or right). The task was self paced, and keypress responses initiated the next
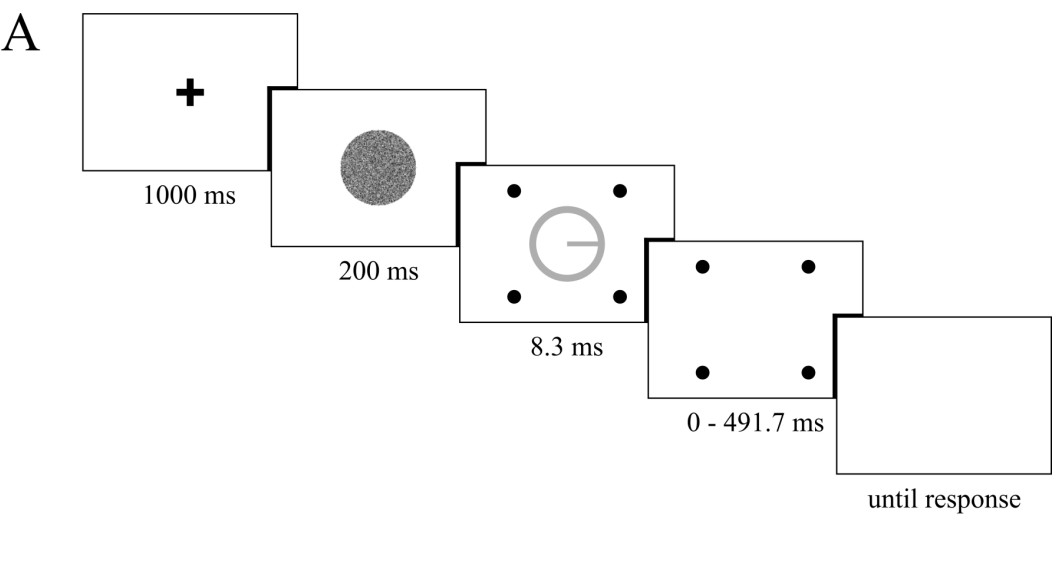

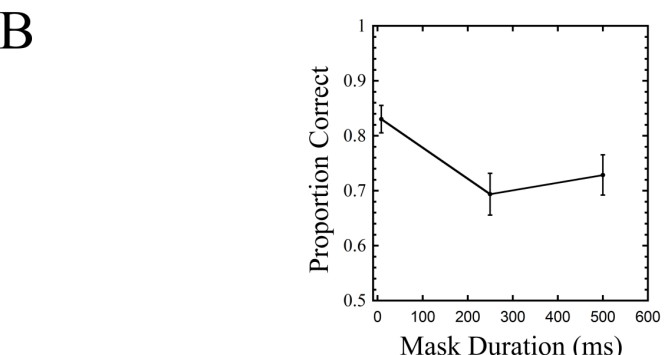

**Figure 2** **Trial sequence and results for Experiment 1.** Stimuli here and in all subsequent figures are rendered in reverse contrast. Error bars indicate standard errors.

trial. Within each run, each of the three mask durations was repeated 30 times in random order, and observers completed three runs, for a total of 90 trials per condition. Seven observers completed this experiment, one of whom is the first author, and the rest of whom were naive. All observers, in this and subsequent experiments, gave verbal consent before participation. This study was approved by the Human Participants Review Committee (HPRC) at York University (approval number HPRC 2014-094). Before beginning these experimental runs, each observer completed a PEST procedure (*Taylor & Creelman, 1967*) for the common offset condition, where the standard deviation of the noise patch was varied until 80 percent performance was achieved. This value was then used in the main experiment for that observer across all conditions. Feedback was not provided in either the PEST phase or the experimental trials.

## Results & discussion

Results are shown in Fig. 2B. A one way repeated measures ANOVA revealed a main effect of mask duration ($F(2, 12) = 4.613$, $p = 0.033$). Follow up comparisons showed a difference between the common offset and 250 ms mask duration conditions ($t(6) = 3.985$,
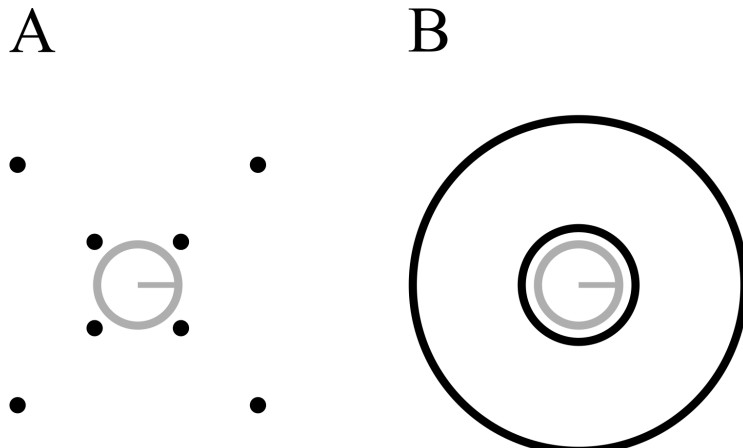

**Figure 3** **Stimuli used in Experiment 2.** (A) Four-dot mask. (B) Annulus mask. Each mask type is shown with both the smallest (7.2 arcmin) and largest (103.5 arcmin) target-mask separations. Stimuli are drawn to scale.

$p = 0.007$), no difference between common offset and 500 ms ($t(6) = 1.942$, $p = 0.1$), and no difference between 250 ms and 500 ms ($t(6) = 0.681, p = 0.521$).

Experiment 1 shows a clear masking effect, with about a 13.5 percent drop in performance between the common offset and 250 ms mask duration conditions. The data do not provide strong evidence for masking at the 500 ms condition, nor do they support recovery from masking (*Goodhew et al.*, *2012*), as there was no difference between the 250 ms and 500 ms conditions. In our next experiment, we measured masking as a function of target-mask separation, with both a 4DM and an annulus mask.

## EXPERIMENT 2

Our next experiment compared a 4DM with an annulus mask as a function of the separation between target and mask. If the annulus mask is acting primarily through local inhibitory mechanisms, while the 4DM involves object-level mechanisms, then as target-mask separation increases, one would expect a drop in annulus masking, while dot masking should remain relatively unchanged.

### Methods & procedures

The general procedure was similar to the previous experiment. Here, runs were blocked according to mask type (4DM, annulus, see Fig. 3). Within each run, four different mask separations were tested, with a constant mask duration of 250 ms (7.2 arcmin, 17.2 arcmin, 27.3 arcmin, and 103.5 arcmin), in addition to a common offset condition for each mask at the lowest separation (7.2 arcmin). Each of these five conditions was tested in random order, 30 times per run, and observers completed three runs for each mask type, for a total of 90 trials per mask type per separation. Nine observers completed this experiment, all of whom were naive. Observers completed runs in alternating order of mask type. Four of them began with the annulus mask, and five began with the 4DM. Instead of running a PEST procedure, observers were trained in the common offset condition, with

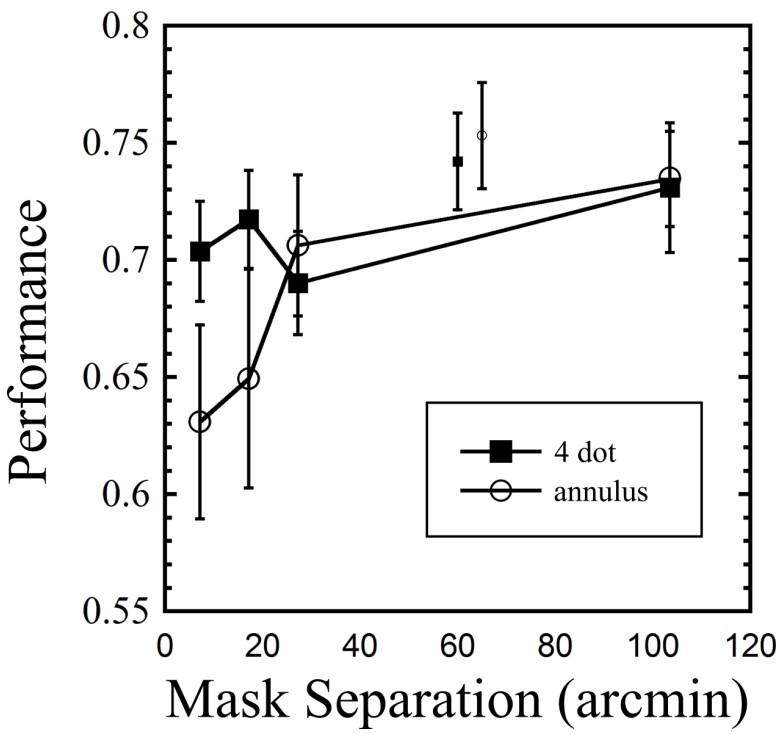

**Figure 4 Results from Experiment 2, showing performance across nine observers for both mask types as a function of separation from target.** The two smaller symbols in the upper middle region indicate baseline values for the two masks (common offset mask, 7.2 arcmin separation). Error bars indicate standard errors.

increasing amounts of noise in the forward mask. This was done until the experimenter had established a noise level at which performance was around 75 percent correct. In these training trials, a correct response generated an auditory tone. In the experimental trials, in this and in all other experiments in this study, no feedback was provided.

## Results & discussion

Results are shown in Fig. 4. A two way repeated measures ANOVA (first factor = mask type: two levels (4DM, annulus); second factor = target-mask separation: four levels, excluding baseline) revealed no main effect of mask type ($F(1, 8) = 1.218, p = 0.3$), a main effect of separation ($F(3, 24) = 7.489, p = 0.001$, and an interaction between mask type and separation ($F(3, 24) = 3.245, p = 0.04$). To verify that masking was obtained with the 4DM, a paired samples $t$-test was used to compare the 7.2 arcmin 250 ms mask with the baseline condition (7.2 arcmin common offset mask), and this confirmed masking ($t(8) = 2.861$, $p = 0.021$). The striking pattern found with the annulus mask is strong evidence of the primacy of spatially local inhibitory mechanisms here. By a separation of 27.3 arcmin, annulus masking was greatly attenuated, compared to the 7.2 arcmin mask ($t(8) = 3.067$, $p = 0.015$). In contrast, the 4DM showed no change in masking between the 7.2 and 27.3 arcmin conditions ($t(8) = 0.77, p = 0.47$).

While other studies have looked at the effect of separating the mask, *as a whole*, from the target (e.g., *Jiang & Chun, 2001*), to our knowledge, only one other study has looked at the

effect of increasing the separation of the individual dots, as was done in our current experiment (*Di Lollo, Enns & Rensink*, *2000*). In that study, masking did not decrease up to the measured separation of 40 arcmin, a finding that closely matches our own. These findings support the idea that (spatially sensitive) local inhibitory mechanisms underlie annulus masking, while (less spatially sensitive) object level mechanisms underlie dot masking.

## EXPERIMENT 3

The two previous experiments had observers select from four possible line orientations. In our final experiment, which only used a 4DM, the task was to adjust the orientation of a reference line until it matched one of 90 possible target orientations. This allowed us to investigate two separate questions. First, are errors more likely to occur when the target line is oriented such that it is closer to one of the four dots? If so, then this would point towards the contribution of spatially local masking mechanisms. Second, how does the distribution of errors change between baseline and masked conditions? Is the representation of a masked target simply less precise (in the orientation domain), or is it never consciously processed, in which case observers would have to guess? Or is it some combination of both? By modelling the errors as a mixture of a Gaussian distribution centred on the actual target orientation (representing trials in which the target was perceived) and a Uniform distribution across all possible orientations (representing trials in which observers were relegated to guessing), we were able to address this question.

### Methods & procedures

The stimuli were similar to the previous experiments; however, the target could now adopt any of 90 unique orientations, spaced at 4° intervals around the circle, ranging from 0° to 356°. Observers used the mouse to adjust and select the orientation of a reference pattern, which appeared 500 ms after target offset. The reference pattern and the target were visually identical, except that the orientation of the target was determined by the current trial, whereas the orientation of the reference pattern was determined by the current mouse position.

Each run contained three conditions. Target alone with no mask, a common offset 4DM, and a 250 ms duration 4DM. The target-mask separation was 10.8 arcmin. Within each run, each condition was tested once at each of the 90 orientations, for a total of 270 trials per run, which were presented in random order. Within each run, a message appeared every 90 trials, indicating the current progress, at which point a mouse click returned the observer to the trial flow. Each observer completed three runs, thus each orientation was tested three times per condition. Nine observers completed this experiment, one of whom is the first author and the rest of whom were naive. Training was done in the common offset condition, and for each observer, the experimenter increased the standard deviation of the noise patch until performance was about 60 percent correct. In this training phase, a correct response was defined as being within 20 degrees of the actual target orientation, and was signalled with a tone.

## Results & discussion

The purpose of this experiment was two-fold. First, we investigated whether there was a bias for increased masking when the orientation of the target line was closer to the four dots. Second, we modeled the distribution of errors for each of the three masking conditions.

For each trial, the difference between the actual target orientation and the reported orientation was calculated. Trials were pooled into two categories: those in which the presented target line orientation was aligned in proximity to the dots (oblique), and those in which it was aligned along the cardinal axes (cardinal), and thus maximally distant from the dots. This classification was done by visual inspection of a modified target pattern, which was identical to the experimental pattern, except that the oriented bar now extended beyond the annulus (Fig. 5A). Upon inspection of each of the 90 bar orientations, if any part of the bar intersected any of the four dots, the orientation was classified as oblique. Cardinal orientations were classified with a similar procedure, but with the four dots arranged in a diamond pattern (see Fig. 5A for a depiction of one of these dots). Of the set of 90 orientations, 18 orientations were classified as oblique, and 18 as cardinal. This meant that for each observer, there were 54 oblique trials and 54 cardinal trials for each of the three masking conditions. Results are shown in Fig. 5B. Note that the errors are unsigned. The overall errors indicate the mean of all 90 orientations.

The first thing to note is that masking did occur. The overall error for the common offset condition was 35.9 degrees, while that of the 250 ms mask was 44.9 degrees, and this difference was significant ($t(8) = 4.122$, $p = 0.003$). Interestingly, the presence of a common offset mask increased performance relative to no mask at all ($t(8) = 3.822$, $p = 0.005$).

If the (trailing) 4DM was exerting its effect through local spatial mechanisms, then a greater increase of errors in the oblique orientations in the 250 ms condition (relative to the common offset baseline) would be expected, compared to the increase in cardinal errors. To test this interaction, we ran a two-way repeated measures ANOVA (first factor = mask condition: two levels (common offset, 250 ms); second factor = target orientation: two levels (oblique, cardinal)). This revealed a strong trend for the effect of mask condition ($F(1,8) = 4.825$, $p = 0.059$; no main effect of error location ($F(1,8) = 3.549$, $p = 0.096$; and no interaction ($F(1,8) = 0.138$, $p = 0.72$).

Our data do not show evidence of an orientation bias in masking: we did not observe increased masking at target orientations proximal to the dot locations, as evidenced by the almost perfectly parallel lines in the right half of Fig. 5B. While this does not definitively prove that dot masking operates via spatially global mechanisms, it is consistent with the finding in Experiment 2 that dot masking is, to a large extent, independent of target-mask separation.

For any trial in which masking occurred, there are at least two possible scenarios. On the one hand, the target may have never been consciously processed, either because the representation was completely obliterated, or because it had been degraded below a critical threshold. In such a scenario, an observer would have to guess the orientation of the target. On the other hand, the target representation may have remained consciously accessible, but in a degraded form. If the nature of this impoverishment was such that the precision

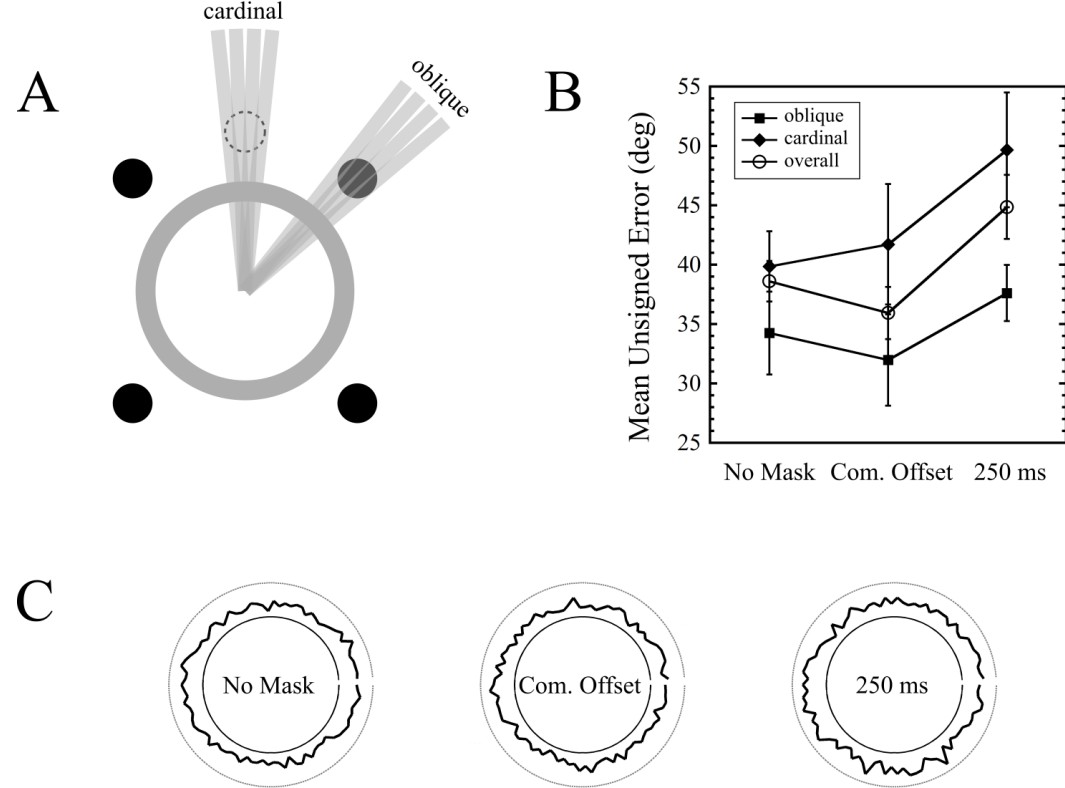

**Figure 5 Orientation bias results from Experiment 4.** (A) Depiction of how oblique and cardinal orientations were defined. The dot shown with the dashed outline served as a marker to classify orientations as cardinal (only one of these dots is shown in the figure). (B) Errors as a function of mask condition and orientation type (oblique vs. cardinal) across nine observers. Error bars indicate standard errors. (C) Polar plots depicting errors as a function of target orientation, for each mask type, across all observers. The magnitude of the error is depicted by the radially projected distance from the inner circle to the thick black curve. The outer circle indicates the error expected by chance performance (90°).

of orientation information was compromised, then we would expect a reduction in the precision of the response. Note that these possibilities are not mutually exclusive. We fit a mixture model to our data to estimate the relative proportions of trials that corresponded to these two possibilities, using a Gaussian and Uniform for the perceived target and non perceived target trials, respectively.

The probability density function for this mixture model, which defines the distribution of signed errors, is defined below in Eq. (1):

$$PDF = W_G * G(\mu, \sigma) + (1 - W_G) * U(-\pi, \pi) \tag{1}$$

where $G$ is a Gaussian function, with mean ($\mu$) and standard deviation ($\sigma$), and $U$ is a Uniform distribution along the specified interval. $W_G$ is the weight of the Gaussian term, and as the mixed distribution sums to unity, the weight of the Uniform term is $1 - W_G$. In the context of our experiment, a lower value of $W_G$ indicates more guessing, and a higher value of $\sigma$ indicates less precise responses of perceived targets. This mixture model is very similar to that used by *Zhang & Luck* (*2008*), except they used a circular normal

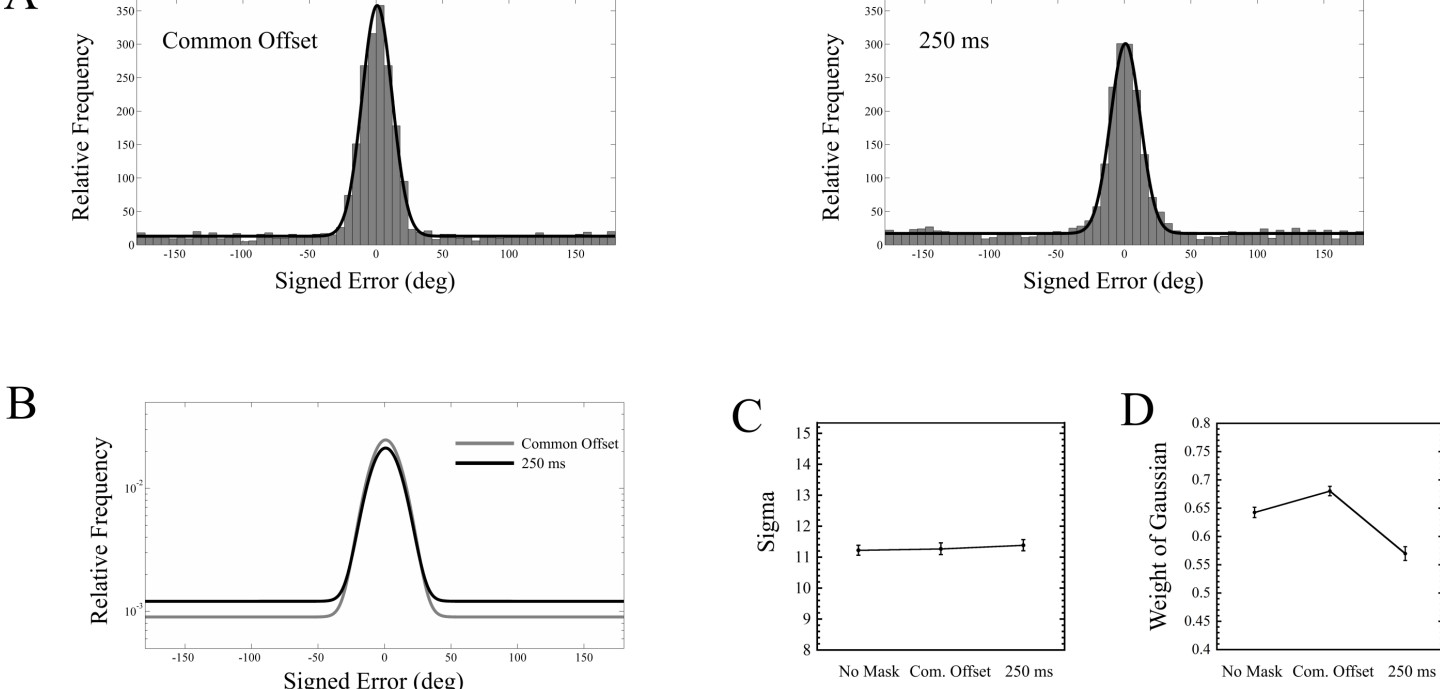

**Figure 6** (A) Distribution of errors across nine observers for the common offset and 250 ms conditions, along with superimposed model fits. (B) Model fits for the error distributions of the common offset mask and 250 ms mask plotted together (note the log scale). Each curve represents a mixture of a Gaussian and Uniform. Note that the width of the Gaussian remains unchanged between these two masking conditions. (C) Standard deviation of Gaussian component for each of the three mask conditions. (D) Weight of Gaussian component for each of the conditions. Error bars indicate standard errors.

distribution, whereas we used a normal distribution. While a circular normal distribution would be a better match for the circular nature of our error variable, which spanned −180°–180°, it has been shown that a single Gaussian is an excellent approximation to a circular normal distribution when the standard deviation is small relative to the range of possible errors (*Shooner et al., 2010*; also see Appendix A in *Agaoglu et al., 2015*).

For each observer, and for each of the three masking conditions (no mask, common offset, 250 ms mask), a maximum likelihood procedure (*Myung, 2003*) was used to estimate $W_G$, $\mu$, and $\sigma$. The results, pooled across all nine observers, are shown in Fig. 6. A one way repeated measures ANOVA was run on each of the two parameters, $\sigma$ and $W_G$, across all three masking conditions. For $\sigma$, there was no main effect of masking condition ($F(2,16) = 0.046$, $p = 0.956$). For $W_G$, a main effect of masking condition was found ($F(2,16) = 12.215$, $p = 0.001$). Interestingly, while the guess rate with the trailing mask was greater than in the common offset condition ($t(8) = 4.664$, $p = 0.002$), it was also higher in the no mask condition than in the common offset condition ($t(8) = 3.985$, $p = 0.004$).

Masking appears to have absolutely no effect upon the precision of responses (Fig. 6C). Instead, the performance drop between the common offset mask and the trailing mask (Fig. 5B) is driven entirely by the target being rendered invisible, where the guessing rate increased from about 32% to 43% (Fig. 6D). This also demonstrates that the

250 ms mask had a measurable impact, relative to the common offset mask, on only a fraction of the trials.

These results are fairly striking, especially when compared against two recent studies that conducted very similar experiments (*Agaoglu et al.*, *2015*; *Harrison, Rajsic & Wilson*, *2016*). In particular, Agaoglu et al. used an oriented bar as a target stimulus very similar to ours, and, using a backward masking paradigm, found that the standard deviation of observers' responses increased from about 11 degrees to as high as 20 degrees, depending upon the type of mask being used (all mask types also increased the guessing rate). On the other hand, the standard deviation of the responses in our experiment remained at around 11 degrees across all masking conditions. *Harrison, Rajsic & Wilson* (*2016*), using a similar masking paradigm, where observers attempted to match the orientation of a gap in a target annulus, found that the standard deviation of responses showed about a 40 percent increase in the trailing vs. common offset condition. We will explore this discrepancy further in the general discussion. While we found no evidence for an effect of masking condition upon response precision, our analysis does not directly say anything about the evidence *against* an effect. To address this, we ran a Bayesian analysis on our data, and have presented this in the Supplemental Information.

## GENERAL DISCUSSION

In Experiment 1, we successfully reproduced four-dot masking in a fully attended and foveated target, as originally reported by *Filmer, Mattingley & Dux* (*2015*). The data from Experiment 2, which directly compared a 4DM with an annulus mask, strongly suggest that these two mask types operate through at least partially distinct mechanisms. Experiment 3 demonstrates that in the context of our stimuli and task, there is no evidence that the encoding precision of the target (in the orientation domain) is affected by masking. Rather, masking appears to be driven exclusively by rendering the target inaccessible to conscious processing. In this section, we will explore these findings in more depth.

As in *Filmer, Mattingley & Dux* (*2015*), we included a forward noise mask in our study. It is an open question, however, whether similar results would obtain had we incorporated the noise mask into the same frame as the target. In the current experiments, while the stimulus onset asynchrony (SOA) of the noise mask, relative to the target, was −200 ms, the onset of the target occurred immediately following the offset of the noise mask (i.e., if we take into account the scanning mode of our display, this would correspond to an interstimulus interval (ISI) of slightly below 8.3 ms). As such, the noise mask and the target would likely be integrated into a unified percept. Inspection of neurophysiological data in *Macknik & Livingstone* (*1998*) suggests that the transient response to the target is slightly lower with an SOA of −100 ms (ISI of 0 ms), relative to a common onset common offset condition (Fig. 4 in their paper). However, the masks in that study comprised a set of flanking bars. A more relevant study (*Agaoglu et al.*, *2015*), which used an almost identical task to ours, showed peak masking with a noise mask with an SOA of 0 ms, which appeared to show even greater masking than that found with a forward mask of SOA −10 ms (ISI = 0 ms). Thus, although we have not tested this, we suspect that the critical element that enables

successful four-dot masking here is the incorporation of noise into the target, rather than the use of a forward mask.

Why would a noise mask be critical for four-dot masking? Its ostensible purpose in *Filmer, Mattingley & Dux* (*2015*) was to reduce baseline performance to below ceiling. Pilot work in our lab, however, showed no evidence of four-dot masking (in the absence of a noise patch) when the contrast of the target was reduced such that baseline performance was well below ceiling. Indeed, if ceiling effects were the only issue here, then it is surprising that it has taken almost two decades to produce reliable four-dot masking in a fully attended and foveated target (although see *Dux et al.*, *2010*). Rather, it is more likely that the target needs to be degraded in a specific manner. With the use of a noise mask, the integrity of the stimulus as a coherent object is dramatically compromised. In order to perceive an oriented bar, perceptual filling in processes may be required, and this may delay or otherwise compromise the formation of a stable object representation, which in turn may leave the target more vulnerable to being dominated by the mask. Another possibility is that if the 4DM is able to disrupt the perception of target elements proximal to the dots, as is the case, for example, in *object trimming* (*Kahan & Enns*, *2010*), this may sometimes leave less information available for the visual system to glean the bar's orientation, if the part of the object being trimmed happens to contain useful information. This would be equivalent to reducing the signal to noise ratio. Both of these possibilities are compatible with the finding in Experiment 3 that suggests when an object is successfully masked here, it is rendered invisible.

It should be noted that while Experiment 1 was very similar to that in *Filmer, Mattingley & Dux* (*2015*), it was not an exact replication, as there were differences in the dimensions and contrast of the stimuli. In particular, while the target and mask in Filmer et al.'s first experiment had the same contrast relative to the background, we used a lower contrast for the target, compared to the mask. Our stimuli are thus more comparable to Filmer et al.'s second experiment, where the contrast of the target was thresholded to the desired performance level.

The data from our second experiment provides evidence for a number of related ideas. First, the finding that dot masking did not attenuate with increasing target-mask separation suggests that four-dot masking operates via mechanisms that are relatively insensitive to spatially local interactions between target and mask. Rather, masking here likely involves mechanisms that primarily involve representations at higher levels of the visual processing hierarchy, where receptive fields are larger (*Dumoulin & Wandell*, *2008*; *Smith et al.*, *2001*). While the involvement of higher visual areas is compatible with object level accounts involving interference between feedback and ongoing input, our data do not directly say anything about whether feedback is involved. Second, the finding that annulus masking *did* attenuate with increasing target-mask separation is strong evidence that here, the mechanisms are sensitive to spatially local interactions between the target and the mask, and confirms the findings of many previous masking studies that have found a similar sensitivity to spatial separation (see § 2.6.6 in *Breitmeyer & Öğmen*, *2006*). Finally, the interaction between mask type and separation adds to the evidence that four-dot masking

operates through mechanisms that are at least partially unique from those involved with masks whose contours fully surround the target.

In our final experiment, we found that masking appears to be driven exclusively by rendering the target completely invisible. While this finding is certainly compatible with the idea of object updating (*Pilling & Gellatly, 2010*), where the original object (target + mask) is updated into a new object (mask), it is not proof of object updating. Another possibility is that four-dot masking simply degrades the target without impacting the encoding precision of its orientation information. For example, if the mask is simply reducing the contrast of the target, then it is possible that orientation encoding is unaffected. *Mareschal & Shapley* (*2004*) found that orientation discrimination of foveally presented gratings was unaffected by contrast when the diameter of these gratings was large (1 degree of visual angle). However, for two out of the three observers in their experiment, contrast did have a dramatic impact on these thresholds when the diameter was as large as 0.5 degrees (all of the observers showed reduced performance as contrast was reduced when the diameter was 0.25 degrees). The length of the oriented bar in our experiment was about 0.5 degrees, and it is certainly possible that our cohort of observers was demonstrating contrast invariance to orientation encoding with this stimulus size. It would be interesting to run our experiment with a bar length of 0.25 degrees and see if precision is similarly unaffected. Indeed, in *Filmer, Mattingley & Dux* (*2015*), the diameter of the target was 0.55 degrees, which meant that the bar length was about 0.25 degrees. Another alternative to object updating is object trimming (*Kahan & Enns, 2010*), where a two dot mask was shown to interfere with target contours that were adjacent to these dots. If object trimming was occurring in our study, then, as discussed earlier, this could account for the target being rendered invisible due to a reduced signal to noise ratio. However, if this were the case, then we might expect more masking when the target orientation was aligned with the oblique axes, and we found no evidence of this (Fig 5B).

Another aspect of Experiment 3 worth considering is that, while in our study, we found no change in response precision, two other similar studies showed a substantial drop in response precision (*Agaoglu et al., 2015*; *Harrison, Rajsic & Wilson, 2016*). However, there are some important differences to consider here. In Harrison et al., the target was an annulus whose gap could adopt a number of different orientations. It is possible that the orientation specific information in such a stimulus may be compromised, while that of an oriented bar would not, given the same general type of mask induced degradation. For example, it is clear how object trimming could result in part of the circle being occluded near the gap, resulting in a larger perceived gap. In such a situation, response precision could certainly suffer. In Agaoglu et al., while the target was very similar to ours (although their oriented bar was almost twice as long as ours), they did not use a four-dot mask. Rather, they compared masking functions between an annulus mask, a noise mask, and a structure mask (the last of which comprised three bars similar to the target bar but in random orientations). It is easy to conceive how, for example, their structure mask could add noise in the orientation encoding domain. Furthermore, the parameter in these masking functions was the SOA of a pulsed mask, rather than the mask duration of a common onset mask. Finally, in both these studies, stimuli were presented non foveally.

In Harrison et al., the target was one among anywhere from two to eight possible targets presented at 3.5 degrees from fixation. In Agaoglu et al., a single target was presented at a 6 degree horizontal eccentricity. While orientation discrimination thresholds do rise as a function of eccentricity (*Sally & Gurnsey*, *2004*), this alone does not explain why precision would be reduced with the addition of a para or metacontrast mask, as shown in Agaoglu et al., or with an increased four-dot mask duration, as in Harrison et al. However, there may be an interaction between contrast and eccentricity. In *Mareschal & Shapley* (*2004*), the effect of contrast upon orientation discrimination thresholds was evident at larger target sizes in the periphery compared to the fovea. When stimuli were presented at 5 degrees of horizontal eccentricity, there was a large effect of contrast upon orientation discrimination of gratings that were as large as 1 degree, whereas in the fovea, all observers showed contrast invariance at this stimulus size. In Agaoglu et al., the oriented bars were 1 degree in length, and were presented at 6 degrees of eccentricity. If their masks were operating through contrast reduction, then this is a plausible explanation of the reduced response precision.

The current study reinforces *Filmer, Mattingley & Dux*'s (*2015*) finding that four-dot masking can be reliably obtained with a fully attended and centrally presented target, an idea which must be accounted for in any theory of masking via object updating. The direct comparison between the two mask types used in our study adds new evidence that four-dot masking operates via unique mechanisms when compared to traditional masking stimuli. Our last experiment provides novel insights into the way in which four-dot masking impacts the visual representation of a foveated target.

On a final note, we have chosen to refrain from making assumptions about the underlying mechanisms of masking, beyond those that can be reasonably supported by our data, and this is reflected in our terminology throughout this article. While our findings are consistent with an object mediated account of masking, we have not introduced any experimental manipulations that test, for example, whether object *updating* (versus object *substitution*) accounts for our masking (*Enns, Lleras & Moore*, *2010*). Accordingly, we have avoided the use of the terms object substitution and object updating wherever possible. Similarly, as our data say nothing about the relationship between feedforward and feedback signals (*Kafaligonul, Breitmeyer & Öğmen*, *2015*), we have avoided implicitly assuming any particular processing regime.

## ACKNOWLEDGEMENTS

We would like to thank the editor and the two reviewers for their time and effort. Their critical insights greatly helped improve upon earlier versions of the manuscript.

### Funding

This research was supported by CIHR Grant #172103 to H.R.W and a grant from the Canadian Institutes for Advanced Research (CIFAR) to H.R.W The funders had no role in study design, data collection and analysis, decision to publish, or preparation of the manuscript.

## Grant Disclosures

The following grant information was disclosed by the authors:
CIHR Grant #172103.
Canadian Institutes for Advanced Research (CIFAR).

## Competing Interests

The authors declare there are no competing interests.

## Author Contributions

- Marwan Daar conceived and designed the experiments, performed the experiments, analyzed the data, wrote the paper, prepared figures and/or tables.
- Hugh R. Wilson conceived and designed the experiments, analyzed the data, contributed reagents/materials/analysis tools, reviewed drafts of the paper, provided guidance throughout the course of the study.

## Human Ethics

The following information was supplied relating to ethical approvals (i.e., approving body and any reference numbers):
Human Participants Review Committee (HPRC)
Approval Number: HPRC 2014-094.

## Data Availability

The raw data has been supplied as Data S1.

## Supplemental Information

Supplemental information for this article can be found online at http://dx.doi.org/10.7717/peerj.2068#supplemental-information.

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
