# Peer review of "A closer look at four-dot masking of a foveated target"

_PeerJ, doi:10.7717/peerj.2068_

## Round 0.1 · original submission · Major Revisions

· Academic Editor

Major Revisions

Both reviewers requested major revisions addressing various feasible improvements and some a few more major concerns about statistical validity and that the "experiments don’t hang together very well".

I agree with the reviewers that these issues should be resolved.

Reviewer 1 ·

Basic reporting

In general, I found the manuscript really interesting, well-written, and thorough. It touches on numerous interesting topics, and provides some interesting data based on very systematic analyses. The relevant details of methods and results are clearly presented.

Experimental design

## Experiment 1

PEST: the staircase was run using only the common offset condition. Was the same contrast then used for all masking conditions?

## Experiment 2

Why was the maximum mask duration so short (83.3) when the strongest mask effect was found for a mask duration of 250 ms in Exp 1?

This experiment had different mask types and different mask durations. The ANOVA yielded no significant effects of neither factor, nor a significant interaction. The analysis should just stop at this point. Certainly, "noting two important features" in the results graph is not enough for computing posthoc tests for each masking duration in the 4DM condition (without correction for multiple comparisons). Likewise, given the insignificant ANOVA results, it is not warranted to go on and test two masking conditions at a selected mask duration.

Most statements about the results in Exp. 2 are not supported by the statistics. For example: "First, and perhaps the most intriguing finding in this experiment, the 4DM did not show a significant drop in performance until a duration of 83.3 ms". Statistically speaking, nothing specific happened at 83.3 in any condition.

Likewise, this statement about Exp. 2 in the General discussion is not supported by the results: "The data from our second experiment suggest rather different masking functions for a 4DM and an annulus mask" – the statistics seem to suggest they are rather the same.

## Experiment 3

The third experiment is not mentioned in the general discussion and it does not connect directly with the other experiments. Is it not relevant enough? In that case, maybe it is not necessary to include this experiment in the paper at all.

# Experiment 4

I did not understand the following part of the methods section: "Classification was done by visual inspection of a modified target pattern, which was identical to the experimental pattern, except that the oriented bar now extended beyond the annulus (Figure 5A). For each of the 90 bar orientations, if any part of the bar intersected any of the four dots, the orientation was classified as oblique."

What exactly was classified here – the orientation of the presented bar? The reported orientation? Why was it necessary to do a visual inspection of these data, aren't they just numbers in the presentation/analysis program?

Please describe how the mixture model compares to other models in the literature, e.g. the mixture model used by Zhang & Luck (2008).

# General Discussion

"Another possibility is that four dot masking simply degrades the target without impacting the encoding precision of its orientation information. For example, if the mask is simply reducing the contrast of the target, then it is possible that orientation encoding is unaffected. Mareschal & Shapley (2004) found that orientation discrimination of foveally presented gratings was unaffected by contrast when the diameter of these gratings was large (1 degree of visual angle)."

I understand how degradation by reduced contrast did not affect orientation precision in the Mareschal & Shapley study. However, as far as I can see, stimuli in that study were still perfectly visible, unlike in the present experiment. I am not sure the result of their study can be compared to the present study, where masking made stimuli invisible.

Validity of the findings

A general shortcoming of this study is that the number of subjects was very small (7, 7, 9, 9) and the number of trials per condition was not particularly large either. As a result, I am afraid this study is severely underpowered. For example, the error bars in the figures are quite large. More importantly, the study yielded several non-significant findings. Some of those are discussed as if they were in fact significant. Other non-significant effects are happily discussed as being non-significant without considering that the sample size just might be too low. Moreover, this discussion interprets the absence of a significant effect as evidence for absence of an effect. The authors might consider increasing their sample size to remedy this point. In any case, statistically insignificant result cannot be discussed as if they were significant. Furthermore, with a sample size so small, it would be more honest to also report, in addition to conventional inferential statistics, the amount of evidence provided by the data for/against the null hypothesis. It might turn out, for example, that the insignificant effect of masking on orientation precision in Exp 4 is inconclusive rather than providing evidence for the absence of an effect.

Additional comments

"Four dot masking is thought to be a powerful demonstration of interactions at a spatially global object level, rather than of spatially local contour interactions (Di Lollo, Enns, & Rensink, 2000)." – Di Lollo et al. have proposed a very specific mechanism: object substitution. The authors seem to be hesitant to use this phrase, maybe because it is a theoretically loaded term. I think this is fine, but it would be interesting to read why they chose not to use this more predominant terminology.

Please report exact p-values (APA style).

Reviewer 2 ·

Basic reporting

Introduction: what was the rationale for comparing four-dot masking with an annulus? The authors state this was a comparison they were interested in, but not why.

Experimental design

Exp 1
o Why reduce the contrast of the target stimulus compared to the dot mask (this was not done by Filmer et al – did pilot work mean it was necessary to keep performance off ceiling?)
o How were the subject numbers chosen? And why do they differ between experiments?
o Why threshold to 80% accuracy?
o For this, and following experiments, do the results change if the author’s data is removed?
Exp 2
o As comparing two different masking conditions, why not threshold performance with no mask, then compare performance under different masking conditions?
o How were the offset times chosen? They don’t correspond to the typical times used in OSM, where masking effects tend to be maximal later (usually ~ 150 – 300 ms). This makes the results quite hard to interpret, as arguably this experiment isn’t really looking at OSM, but some form of earlier process. This is supported by the much smaller masking effect found for the four-dot mask compared to exp 1.
o Why was mask type blocked, and not randomized on a trial by trial basis? Or at least, why were mask type blocks not interleaved?
o Back to the timings of the mask, I don’t think the masking functions are necessarily dissociable. It just looks like the annulus is creating a larger masking effect for these early mask offset conditions.
Exp 3
o Why use different thresholding level, and procedure?
o Was feedback provided in Exps 1 and 2?
o Having found a difference in the mask with relatively short duration masks in Exp 2, why now use a different mask duration in Exp 3? It certainly makes more sense to go with ~ 250ms, but I’m not sure of the rational for keep changing these parameters.
Exp 4
o Again, why use a different level of thresholded performance?
o How many participants?
o Could the increased precision for the common onset condition, relatively to delayed offset and no mask, be due to the training/thresholding received for this condition alone?

Validity of the findings

No Comments.

Additional comments

Daar & Wilson begin by running a replication of the Filmer et al (2015) study. This is a great way to start the paper. Nice topic as well, to investigate different masking conditions with foveal presentation. I like the paper and the experiments, but I am left with a feeling that the experiments don’t hang together very well. Multiple changes were made between each experiment, and the result is they are difficult to compare and contrast and leave me unsure of the conclusions. This could be mitigated by a clear rational/explanation for each of the changes.

---

## Round 0.2 · Minor Revisions

· Academic Editor

Minor Revisions

There remains an issue about the statistical interpretation. Please address this.

Reviewer 1 ·

Basic reporting

ok

Experimental design

ok

Validity of the findings

I have one remaining concern, which was in my opinion not adequately addressed. As I noted previously, the number of subjects is quite small (7 and 9). This is a severe limitation for inferential statistics. At the same time, the number of trials per subject and condition is not large enough for the classical psychophysical approach of showing results of single subjects with thousands (instead of 90) of trials each instead or inferential stats. Moreover, the authors interpret some of the statistically insignificant effects as evidence for the null hypothesis (e.g. that masking has no effect on the circular standard deviation). Even though this interpretation may appear reasonable, it is not warranted in the null-hypothesis-testing framework. In my previous review, I already recommended that the authors report the amount of evidence for the H1/H0. I guess I should have been more explicit. I specifically recommend that the authors report, in addition to t-tests/ANOVAs, a Bayesian quantification of evidence for the H1/H0. Such measures are very easy and straightforward to obtain with freely available software, for example with JASP: https://jasp-stats.org/

Doing so would offer the authors and readers two great advantages: (1) they could substantiate the interpretation that there is positive evidence for their hypothesis, rather than inconclusive evidence. This would be very valuable regarding the small sample size. (2) it would allow a direct quantification of evidence for the H0, whenever this is theoretically interesting (e.g. for the effect of masking on standard deviation).

Additional comments

The authors have replied to most of my previous concerns to my satisfaction. The revisions have strengthened my impression that this is a good paper, which will receive a lot of attention in the field of masking research.

I have one remaining concern, which was in my opinion not adequately addressed (see above).

Reviewer 2 ·

Basic reporting

• Figures seem a bit low res in my copy of the paper.

Experimental design

• Thresholding. The authors explained the rational for the changes in procedure and accuracy levels between experiments in their response to reviewers. However, I have two comments relating to this. First, they do not explain these differences in the paper. This is important for the reader to understand why these changes were made. Second, I’m still not sure of their rationale for changing the thresholded level of performance. Of course, the task does change between experiments (especially from exps 1&2 to 3), and this does presumably change the level of ‘floor’ performance for the task. But why change from 75-80% to 60% accuracy threshold? Surely, for a masking paradigm, there is still the greatest chance of finding an effect when performance is just off ceiling (maximal level of disruption, before floor, remains possible)? If the authors have a different opinion/rationale, please explain it (and mention this in the paper).
• Participant numbers. The choice of participant numbers between experiments still seems relatively arbitrary to me. If a power analysis was conducted, this should be included in the paper (and strengthens the authors arguments). The numbers are still very much on the low side, and I can’t help feeling this may account for some of the differences in masking magnitude for the four-dot mask (especially between Exp 1 and 2).
• Experiment 2. The results from this experiment look like the annulus mask is simply having a larger effect (in terms of mask separation). Thus could the differences be quantitative and not qualitative? After all, the annulus mask does cover a larger area of the screen – there is essentially more of it. So a large masking effect, but with the same magnitude, is a distinct possibility. Can the authors discuss this in the paper (or run a control experiment to rule this out)?

Validity of the findings

• Discussion. The authors mention pilot work indicating that there may be a certain type of degrading of the target representation that is critical for finding foveal four-dot masking, and spend a fair amount of the discussion on this topic. I find this inappropriate given that the data have not been included in the paper, or published elsewhere. The topic is certainly very interesting, and I would encourage the authors to either remove this paragraph and publish separately on this issue, or include the relevant data in this paper.

Additional comments

None.

---

## Round 0.3 · accepted · Accept

· Academic Editor

Accept

The statistical concern raised before was sufficiently addressed in the peer review.